# Psychometric Properties of the European Portuguese Version of the Memorial Emergency Department Fall Risk Assessment Tool

**DOI:** 10.3390/healthcare10030452

**Published:** 2022-02-28

**Authors:** Maria dos Anjos Coelho Rodrigues Dixe, Ana Querido, Susana Mendonça, Pedro Sousa, Helena Monteiro, Daniel Carvalho, Paulo Lopes, Pedro Rodrigues

**Affiliations:** 1Center for Innovative Care and Health Technology (ciTechcare), Polytechnic of Leiria, 2411-901 Leiria, Portugal; ana.querido@ipleiria.pt (A.Q.); susana.mendonca@ipleiria.pt (S.M.); pmlsousa@esenfc.pt (P.S.); 2Nursing School of Coimbra, Health Sciences Research Unit: Nursing, 3046-851 Coimbra, Portugal; 3Hospital da Figueira da Foz, 3094-001 Figueira da Foz, Portugal; lenafmonteiro@gmail.com; 4Centro Hospitalar de Leiria, Rua das Olhalvas, 2410-197 Leiria, Portugal; daniel.carvalho@chleiria.min-saude.pt (D.C.); paulo.lopes@chleiria.min-saude.pt (P.L.); pedro.rodrigues@chleiria.min-saude.pt (P.R.)

**Keywords:** accidental falls, adult, hospital emergency service, validation study

## Abstract

Falls are a public health problem that cause serious damage to people’s health and health systems. This study aims to estimate the validity and reliability of the Memorial Emergency Department Fall Risk Assessment Tool for the European Portuguese population. The sample included 186 adults from an emergency department of a District Hospital in Portugal. Reliability and precision (inter-rater reliability) are assessed by two independent raters. The relationship between MEDFRAT and the Morse Fall Risk Scale is evaluated. All items presented a high Kappa index. The MEDFRAT showed a high and significant correlation with the Morse Fall Risk Scale. The influence of sociodemographic and clinical data was also checked. The MEDFRAT is adequate, valid and reliable for the European Portuguese population to assess the risk of falling of emergency department patients.

## 1. Introduction

The falls of patients in healthcare services are a worldwide concern as they lead to temporary or permanent disabilities [1], decreasing their quality of life, independence and well-being, and are among the most frequent reasons of secondary injuries in hospitals [2]. Patient falls account for almost 40% of adverse events and occur during 7% of hospital admissions [3] and falls are the leading cause of fatal and non-fatal injuries among older adults [4]. 

Falls in hospitalized patients are one of the most common adverse events, and there is an increasing number of studies and data on inpatient falls, but there is limited published evidence on falls in emergency departments (EDs) [5,6,7]. Although the lack of evidence, in EDs, falls occur in younger users, conditioned by factors associated with alcohol or other illicit substances and mental status changes, and are more likely to occur in the first hours after admission to the ED [5].

The ability to accurately and quickly identify patients at high risk of falling in the ED is the most important step in preventing falls and avoiding their consequences. Given the above, it is important to have a valid and adequate tool to assess the fall risk of ED patients.

A fall is defined as an involuntary and incorrigible displacement from an individual’s initial position to a lower level, resulting from the complex interaction of multifactorial circumstances, which may be intrinsic, extrinsic, behavioral, and environmental [8]. Falls are considered a public health issue worldwide, and, in hospitals, they are responsible for two out of every five adverse events related to patient care [9,10]. McErlean and Hughes [5] also refer that falls are a significant cause of morbidity and mortality in healthcare institutions, increasing the number of days of hospitalization and the increase in healthcare costs.

Of the injuries resulting from falls, 2.9% to 9.2% are classified as serious injuries and ≤ 1.8% result in the death of the afflicted individual. Naturally, these events lead, in most cases, to patients staying longer in hospital and additional diagnostic tests being performed, which raise costs that are unnecessary if preventive measures are not applied [11].

The risk factors for falls among patients in the ED are different from the risk profile for falls among patients in other health care services, namely the type of medication. The assessment should be performed upon admission to the hospital or ED, since this information is important for planning nursing interventions. In this sense, it would allow nurses, who are responsible for care systematization, to perform the inherent care to avoid adverse events [12,13].

A recent systematic review of the literature [7] found several instruments that assess the risk of falls, and considers age, cognitive status, general health status, comorbidities, hospital, or home context as some of the important characteristics. In addition to using the correct scale, there is a need for the adequate training of professionals [13].

In a retrospective review, Terrell et al. [3] found that the used instruments had low sensitivity to assess the risk of falls in ED patients. Only 37.5% of patients who fell were identified as having a high risk of falling. On the other hand, McCarty et al. [14] concluded that there was a need not only for a specific instrument for fall risk identification and prevention, but also for the training of nurses (Iowa Model).

The Institute for Emergency Nursing Research [15] and Joint Commission International for Accreditation Standards for Hospital [7] validated the need for an evidence-based ED-specific fall risk assessment tool to assist nurses in customizing preventive interventions related to ED patients’ fall risk. The principal aim of using a fall risk assessment tool is to identify the individual at high and low risk [16].

The scale used in many countries, including Portugal, is the Morse scale. This scale was developed to be applied in acute situations, and for users in rehabilitation and in homes; however, it is ineffective and inefficient in capturing a patient’s risk of falling in EDs [7,17,18].

Two scales were identified to assess fall risk in emergency department patients and to improve the quality of care: The Memorial Emergency Department Fall Risk Assessment Tool [15] and the KINDER 1-Fall-Risk Assessment Tool [17]. These instruments were specifically designed for the rapid identification of patients at risk of falling, as well as for the reassessment of patients for fall risk during their stay in the emergency department.

The existence of only these two specific scales to assess the risk of falls in the context of an ED was confirmed in a recent systematic review of the literature [17].

Both scales are recent and require further validation studies [19]. The KINDER 1 fall risk assessment was designed for use in EDs to rapidly and accurately identify a patient’s risk of falling, but required further testing to ensure validity [17].

Similar to what happened in another study [20], a group of expert nurses working in EDs analyzed the two scales as well as the existing literature, considering that, taking into account the Portuguese reality, MEDFRAT would be the one that best meets their needs.

Among these instruments, the Memorial Emergency Department Fall-Risk Assessment Tool (MEDFRAT) proved to be the most appropriate for users in an ED setting [14,20] with a high ability to predict the fall, requiring little time for its use [20]. The absence of a Portuguese version of the MEDFRAT justifies the present study.

Therefore, this study aims: to perform the cultural validation of the European Portuguese version of MEDFRAT; to determine the psychometric characteristics of the MEDFRAT; to assess the risk of falling in adult and elderly ED patients; and determine the relationship between fall risk and socio-demographic and clinical variables.

## 2. Materials and Methods

### 2.1. Population and Sample

The target population was composed of adult patients who met all the following criteria: attended the emergency department (ED) and aged between 18 and 65 years. Based on the inclusion criteria, the non-probability accidental sample was composed of 186 adult and elderly patients who attended the ED of a Portuguese District Hospital between March and July 2020. Patients who needed an emergency intervention were excluded.

### 2.2. Instruments

Data were collected by nurses working in the service using two data collection methods: case file consultation and observation. The nurses had at least a Bachelor’s degree in Nursing, and all had experience in the ED. The collected data were:

Sociodemographic and clinical data (age, sex, diagnosis/cause for coming to the ED, blood pressure value, medication taken at home, medication taken in the ED, history of falling in previous hospitalizations, history of falling during this hospitalization, and whether he/she was accompanied by family members).

Degree of dependence assessed through the Barthel Dependence Scale. This consists of ten questions with a score ranging from 0 to 20, with lower scores characterized by lower functionality and higher scores by higher functionality [21].

Morse Fall Risk Scale. There are six criteria to assess the risk of falling: history of falling in this hospitalization/urgency or in the past three months; secondary diagnosis; assistance in ambulation (none/nursing/waiter/wheelchair assistance, Crutches/cane/cane/walker); leans on furniture to walk; intravenous therapy; posture while walking and transferring; mental status. The total scale rating is between 0 and 125 points, where users depending on the score are at risk of falling as follows: no risk (0–24 points), low risk (25–50 points) or high risk (≥51 points) [22].

MEDFRAT. This scale consists of six indicators: history of fall in the past 3 months, including since admission with 4 response options; confusion/confusion or disorientation/disoriented with two response options; intoxicated or sedated with two response options; gait alteration with two response options; use of mobility support devices with two response options; altered elimination with two response options [15].

The MEDFRAT score ranges from 0 to 14. After calculating the score for all risk factors, the results were categorized into: (a) low fall risk, if it results in 1 to 2 points; (b) moderate, if 3 to 4 points; (c) high, if more than 5 points. The original assessment of interrater reliability demonstrated 100% agreement between raters.

The instrument translation process was carried out according to international guidelines [23], starting with the authors’ permission of use. The translation from English to Portuguese was carried out by three of the authors of this research, who are fluent in English. After the translations were analyzed and the consensus version was obtained, the back-translation was performed by two professional English translators. After the results of the back-translation (carried out by translators from Portuguese to English), the harmonization of all versions was performed to detect and deal with any discrepancies that might arise between different language versions, ensuring conceptual equivalence.

Content and face validity was submitted to a Delphi panel consisting of 8 nurses (with a Master’s degree and experience working in the falls area) and 3 experts in scale validation (with a PhD degree and experience in the field of falls). The level of agreement for each item was 100%. Each of the Delphi panel members gave their opinion on the adequacy of the translation and the relevance of the item.

The pre-test was conducted by 2 nurses who collected data from 20 users and there was no difficulty in using the instrument. The final version was revised and then used for psychometric testing.

### 2.3. Formal and Ethical Procedures

The project was approved by the institution and had a positive opinion from the Ethics Committee (CE-Nº 31/18). Data were collected separately and independently by two nurses, so that there was no manipulation in the answers to the questions. Before data collection, the nurses who participated in data collection were trained and given a detailed explanation about the scales and how to complete them. The Morse Scale and the MEDFRAT were applied to voluntary users who signed the informed consent form.

The users who were not conscious and guided to sign the informed consent were asked to sign in the presence of their legal representative.

### 2.4. Data Processing

Data were organized and analyzed using the Statistical Package for the Social Science (SPSS) computer program.

For the validation of a scale, it was necessary to ensure that the test measures what it intends to measure, i.e., validity [21,24]. Thus, to achieve this objective, statistical tests were used as follows: (i) Kappa coefficient to assess the level of agreement between the two observers regarding the indicators of the MEDFRAT; (ii) Pearson coefficient to assess the relationship between the two scales; and (iii) Student’s t-test to assess the mean differences between two variables.

## 3. Results

### 3.1. Sociodemographic and Clinical Characteristics

Females were prevalent with 54.8% (102) compared to males with a percentage of 45.2% (84). There are multiple causes for coming to the ED; however, the most frequent are: abdominal pain (19.4%); low back pain (12.4%); difficulty breathing (5.9%); headache (5.9%); diarrhea/vomiting (5.4%); precordial pain (8.1%); pain in the lower limb (4.8%); pleuritic pain (4.3%); melena/retorrhagia (4.3%); and other causes (24.1%).

Most users with a percentage of 74.2% (138) went to the ED accompanied by family members, unlike the remaining 25.8% (48) who went alone.

With regard to taking medication, we found that 36% of users did not take any medication at home, 14% of users took one medication, and 11.8% took four medications, with 41 (22%) users taking five or more medications.

Concerning the medications that users took at home, antihypertensives are the most frequent (40.3%), followed by anti hypercholesterolemic (23.7%), diuretics (10.8%), and anticoagulants (14.5%). The remaining percentages are distributed among several drugs.

Regarding the medication administered to users in the ED, we found that 30.6% are medicated with two drugs, 28.5% with one drug, 21% with three drugs, 8.1% with four drugs, and 3.7% were administered more than four drugs (Table 1).

In hospital, the medication administered with the highest percentage is distributed as follows: analgesics (47.8%), anxiolytics (22.6%) and gastric protectants/anti-inflammatories (19.9%).

Before hospitalization, 185 (99.5%) users had not suffered falls.

Regarding the areas of dependence of the 186 patients who went to the ED, they were mostly independent in all domains, with an average degree of dependence of 82 ± 18.7 (Table 2).

### 3.2. Determination of the Psychometric Characteristics of the Scale

Table 3 shows a high level of agreement between the two observers in all six indicators of the MEDFRAT (*p* < 0.001). It is worth noting the existence of a high value of sensitivity and specificity.

In this assessment, some risk factors are scored higher than others, such as: if the patient is confused or disoriented, 5 points are scored; if the patient is intoxicated or sedated, 3 points are scored; if the patient uses a walking aid, 1 point is scored; and if the patient has altered elimination, 1 point is also scored. Regarding the history of falling within the past 3 months, the score differs according to the type of fall: a single mechanical fall is scored 1 point, a physiological fall is scored 2 points, and if he/she is prone to fall he/she is scored 3 points.

When assessing the concurrent validity with the Morse fall risk scale, we verified that, for both observers, the correlation value is positive, moderate and significant, denoting that both instruments assess the same construct (Table 4).

Considering the overall value presented in the scale (1 ± 2.1 observer 1 and 1 ± 2.1 observer 2), we can say that the fall risk of the users in our sample is low. It should be noted that there are no statistically significant differences between the assessments made by the two researchers (*p* > 0.05).

Since observer 1 is considered the gold standard (the principal investigator, given his training and trinity, was considered golden standard) and there are no statistically significant differences between the evaluations of the two evaluators, we chose to present data from researcher 1 only as they refer to the data from the gold standard observer.

Regarding the assessment of the risk of falling, in Table 5, we can see the distribution of the sample according to the risk of falling.

### 3.3. Relationship between Fall Risk and Sociodemographic and Clinical Variables

By analyzing Table 6, we can see that the risk of falling increased with age, number of medications and decreased with increasing blood pressure, and with increasing dependence level.

There were no differences in the risk of falling with the patient’s gender (*t* = 1.631; *p* = 0.104).

## 4. Discussion

The main purpose of this study was to validate the Memorial Emergency Department Fall Risk Assessment Tool (MEDFRAT) into European Portuguese. This scale was intended to be validated as the most appropriate to the context of care in the emergency department.

Data collection of 186 patients was performed by two observers, as in a study by Scott et al. (2018). The observers (nurses) were educated and trained about the applicability of the scale in the week before data collection began. In this study, data collection took place over 5 months, a slightly longer period than in the original study [25]. This longer period was due to the pandemic situation that occurred during the study period. Another point in common between these two studies was the use of two nurses with extensive experience in the ED, as well as the fact that the sample of users was older than 18 years of age. Regarding the data collection schedules, both in this study and the study of Scott et al. [25], data were collected during daytime shifts.

In the study by Scott et al. [22], the sample had ages distributed between 21–40 and 41–60 years old; the majority were male (57%), with female (54.8%) users predominating in this study. Additionally, of note is the study by Horbach et al. (2019) in which the prevalence of males and females was equal. The predominant causes for coming to the ED were abdominal pain, low back pain, precordial pain and breathing difficulty and headache (with the same percentage), while in the study of Scott et al. [25] were altered state of consciousness, chest pain, abdominal pain and intoxication.

Regarding the inter-observer agreement rate, the Kappa statistic was used to assess the degree to which two or more observers agree in classifying the data. According to Scott et al. [25], Kappa investigates the possibility of an agreement by chance versus agreement due to complete reliability. Thus, K = 1 signals complete agreement/reliability and K = 0 designates purely coincidental agreement. The study by Scott et al. [25] shows a K = 0.70 and is considered an acceptable level of agreement. In our study, we can see that there is a high level of agreement between the two observers in all six indicators of the MEDFRAT. In this study, the value obtained in the Morse Scale is 0.979 and in the MEDFRAT is 0.948, both with *p*-values < 0.001, so it can be stated that the coefficient is excellent.

Regarding the two scales, Morse’s and MEDFRAT, we found that the former is more directed to inpatient settings with six items (where each item may have two or three options to assess the inpatient, mainly in walking aid, walking posture, transfer and mental status). The MEDFRAT is more suitable for the ED since it has one item for the various options of falling and whether the patient is intoxicated or sedated. It should also be noted that both scales have a moderate correlation between them (values ranging from 0.475 to 0.509).

According to the assessment with the MEDFRAT, 33.3% of Portuguese users had a risk of falling, with 18.8% being low risk, 7.5% moderate risk and 7% high risk, values lower than those found in other studies [25,26] and similar to those found by Southerland et al. [27]. The fact that our sample had a fall risk of 31.7% and in Scott et al.’s [25] study the fall risk was 53% makes us wonder why there is such a difference. In any case, in the Scott et al. [25] study, one of the causes of coming to the ED was altered consciousness.

Another finding that resulted from the original study [25] was the importance of training for fall risk prevention, since they found that, after 1 year of applying the MEDFRAT and undertaking training, there was a significant decrease in the fall rate of 48% in the ED (from 1.17 to 0.57 falls per 1000 users/day). In this study, due to limitations in data collection caused by the pandemic, it was not possible to assess whether the risk of falling at the service entrance decreased or not, as well as to determine the predictive nature of the scale.

There were no differences between fall risk and participants’ gender (*p* < 0.05) as found in other studies [28,29,30]. The same results were not reached by other authors [3,31] who found that men were more likely to fall than women.

In the literature, the risk of falling increases with age [30,32], as seen in this study. However, it should be noted that, in the retrospective study by other authors [5], it was found that users who fell in the ED were younger than users who fell in other services and that the prevalence of falls was associated with alcohol and substance use.

It is known that having a chronic disease increases the risk of falling, particularly in patients diagnosed with hypertension, diabetes and cardiovascular diseases. In this study, having low blood pressure values is a risk factor for an increased risk of falling, as in other studies [30]. Patients with mobility and balance disorders have a higher risk of falling [30]. Although this study did not assess balance, but rather the level of independence, we can confirm that users with a higher level of independence have a lower risk of falling.

The more medications users take, the greater the risk of falling, and the drugs most associated with the risk of falls are sedatives, hypnotics, antidepressants, benzodiazepines antiarrhythmics, antiepileptics [33].

The fact that the settings where the studies were conducted are different, as well as the use of different assessment instruments for fall risk assessment, means that comparisons of results have to be carefully analyzed.

## 5. Conclusions

This study showed that there are no significant differences between female and male fall risk. It was also found that the Memorial Emergency Department Fall Risk Assessment Tool is valid and reliable for the Portuguese population, taking into account the results of the statistical tests performed. The almost perfect agreement between observers regarding the application of this scale should be highlighted. Thus, the sample allowed us to confirm that the MEDFRAT is adequate to be applied in the emergency department.

The limitations of the study are related to the sample size resulting from the difficulty in collecting the questionnaire due to the SARS-COV-2 pandemic situation. In the future, it will be important to conduct a study with a more representative sample in terms of the number and geographical distribution of users, to make the instrument more psychometrically robust. It would also be important for the instrument to be validated for other cultures to facilitate data comparability.

Another limitation of the study is related to the moments in which the data were collected, that is, on the days when the trained nurses were available to collect them, and data were not collected from all the people who met the inclusion criteria.

For future studies, it would be important to assess the risk of falling on admission and throughout hospitalization so that it is possible to evaluate the predictive power for falling.

## Figures and Tables

**Table 1 healthcare-10-00452-t001:** Distribution of sample responses regarding sociodemographic and clinical characteristics and history of falls.

				Home		Hospital	
		n	%	n	%	n	%
Sex	Female	102	54.8				
	Male	84	45.2				
Cause of coming to the Emergency Department	Abdominal pain	36	19.4				
	Low back pain	23	12.4				
	Difficulty breathing	11	5.9				
	Headache	11	5.9				
	Diarrhea/vomiting	10	5.4				
	Precordial pain	15	8.1				
	Pain in the lower limb	9	4.8				
	Pleuritic pain	8	4.3				
	Melena/retorrhagia	8	4.3				
	Hematuria	4	2.2				
	Anxiety	6	2.2				
	Other causes	45	24.1				
Fall history	Yes	1	0.5				
	No	185	99.5				
Family follow-up	Yes	138	74.2				
	No	48	25.8				
Number of medications	0			67	36	15	8.1
	1			26	14	53	28.5
	2			15	8.1	57	30.6
	3			15	8.1	39	21
	4			22	11.8	15	8.1
	>4					7	3.7
	≥5			41	22		

**Table 2 healthcare-10-00452-t002:** Distribution of the sample responses as to the type of dependency by area of dependency.

	1		2		3		4		5	
	n	%	n	%	n	%	n	%	n	%
Feeding			11	5.9	175	94.1				
Dressing	10	5.4	14	7.5	162	87.1				
Bathing	27	14.5			159	85.4				
Grooming	24	12.9			162	87.1				
Toilet use			24	12.9	162	87.1				
Bowel control					173	93.0	4	2.2	9	4.8
Bladder control					172	92.5	12	6.5	2	1.1
Climbing stairs	12	6.5	18	9.7	156	83.9				
Transfer	9	4.8	30	16.1	147	79.0				
Walking	8	4.3	20	10.8	158	84.9				

(1) Dependent; (2) Needs assistance; (3) Independent; (4) Fecal incontinence and/or with urinary catheter, (5) Occasional incontinence.

**Table 3 healthcare-10-00452-t003:** Results of the level of agreement of the Kappa Coefficient between the two observers regarding the indicators of the MEDFRAT.

		0	1	2	3	5	Kappa Index	*p*
		%	%	%	%	%		
0	Altered elimination (Obs. 1)	98.7	1.3				0.877	0.000
	Altered elimination (Obs. 2)	97.5	7.1					
1	Altered elimination (Obs. 1)	13.3	86.7					
	Altered elimination (Obs. 2)	2.5	92.9					
0	Mobility support devices (Obs. 1)	97.5	2.5				0.877	0.000
	Mobility support devices (Obs. 2)	98.7	13.3					
1	Mobility support devices (Obs. 1)	7.1	92.9					
	Mobility support devices (Obs. 2)	1.3	86.7					
0	Gait change (Obs. 1)	98.7	1.3				0.930	0.000
	Gait change (Obs. 2)	98.7	5.7					
1	Gait change (Obs. 1)	5.7	94.3					
	Gait change (Obs. 2)	1.3	94.0					
0	Intoxicado/sedado (Obs. 1)	100				0.0	1.000	0.000
	Intoxicado/sedado (Obs. 2)	100				0.0		
1	Intoxicado/sedado (Obs. 1)	0.0				100		
	Intoxicado/sedado(Obs. 2)	0.0				100		
0	Confused/disoriented (Obs. 1)	98.9	0.6			0.6	0.823	0.000
	Confused/disoriented (Obs. 2)	98.9	100			9.1		
1	Confused/disoriented (Obs. 1)	100	0.0			0.0		
	Confused/disoriented (Obs. 2)	0.6	0.0			0.0		
5	Confused/disoriented (Obs. 1)	9.1	0.0			90.9		
	Confused/disoriented (Obs. 2)	0.6	0.0			90.9		
0	Fall history in the last 3 months (Obs. 1)	99.4	0.6	0.0	0.0		0.943	0.000
	Fall history in the last 3 months (Obs. 2)	99.4	9.1	0.0	0.0			
1	Fall history in the last 3 months (Obs. 1)	9.1	90.9	0.0	0.0			
	Fall history in the last 3 months (Obs. 2)	0.6	90.9	0.0	0.9			
3	Fall history in the last 3 months (Obs. 1)	0.0	0.0	0.0	100			

(Obs. 1): observer 1; (Obs. 2): Observer 2.

**Table 4 healthcare-10-00452-t004:** Pearson’s correlation between Morse’s scale and MEDFRAT.

	Morse Scale (Observer 1)		Morse Scale (Observer 2)	
	r	*p*	r	*p*
MEDRAFT (Observer 1)	0.487 **	0.000	0.509 **	0.000
MEDRAFT (Observer 2)	0.475 **	0.000	0.503 **	0.000

**Table 5 healthcare-10-00452-t005:** Characterization of the sample regarding the risk of falling in the MEDFRAT.

Fall Risk (MEDFRAT)	n	%
No Risk	124	66.7
Low Risk	35	18.8
Moderate Risk	14	7.5
High Risk	13	7.0

**Table 6 healthcare-10-00452-t006:** Relationship between fall risk and age, maximum blood pressure, minimum blood pressure, degree of dependency and number of medications.

	r	*p*
Age	0.290	0.000
Maximum BP	−0.179	0.015
Minimum BP	−0.148 *	0.044
Degree of dependency	−0.654 **	0.000
Number of medications	0.247 **	0.001

* Significant. ** Very significant.

## Data Availability

All data are available from the corresponding author upon reasonable request.

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
