# Peer review of "Psychometric Properties of the European Portuguese Version of the Memorial Emergency Department Fall Risk Assessment Tool"

_healthcare, 2022, doi:10.3390/healthcare10030452_

Round 1

Reviewer 1 Report

Overall, the manuscript is interesting and worth to be published. But before the publication, it is necessary to be edited by the following suggestions.

In the Introduction, there is not sufficient literature to justify and support the need of the study. Especially, Falls-related literature needs to be added more. In the current format, the authors do not successfully justify the use of MEDFRAT. The authors need to provide more contents of the falls-related instruments and the limitations of the previous tools. Then more comprehensive and persuasive contents of why MEDFRAT is superior and worth to be translated, compared to other instruments.

In the method, the table of the descriptive analysis of translated items, including values of mean, standard deviation, skew, and kurt, as well as the full description of the items. Plus, confirmatory factor analysis and AVE calculation need to be conducted.

In the discussion, some contents need to be moved to the result section. Plus overall, the implications of the results need to be more provided. For example, in line 277, the authors talked about Terrell et al.’s study is different from this study, but the authors do not provide the implications of the difference. Same suggestions can be applied to line 292.

Overall, there are too many short paragraphs. The authors need to combine some of them. For example, the lines 45 and 46 can be edited, the section 3.1 can be edited, and the lines 263 and 264 can be edited.

Line 17. Toll->tool

Author Response

thank you very much for your comments, please check the attachment
They have enabled us to improve the article
The changes are marked in the article
The following changes were not made: in the method, the table of the descriptive analysis of translated items, including values of mean, standard deviation, skew, and kurt, as well as the full description of the items. Plus, confirmatory factor analysis and AVE calculation need to be conducted.
These were not performed because some of the items are dichotomous, which is not recommended by several authors 
we are available to introduce other suggestions that may be pertinent and suggested 

Reviewer 2 Report

The study by dos Anjos Dixe and colleagues describes the assessment and validation of a Portuguese version of an ED Fall-risk assessment tool, the MEDFRAT. The study is thorough, well described and carried out and addresses the important issue of reducing adverse events (e.g. Falls) in ED patients.
I recommend the excellent and detailed work that went into proper translation of the tool and the subsequent assessment and validation of this translated version.

I have only a few minor comment:
While table 5 provides the demographic variables that were found to be related to fall risk, there should be a table stating the demographics of the study population.
In line 160 it states that 74% of patients were accompanied by family members, the remaining 25% were thus arriving at the ED alone? Or accompanied by a non-family member? 
The paragraph starting at line 237 is comparing the demographics of the present study with that of the study by Scott. It should be rephrased to make it clearer which values belong to which study (Scott or the present manuscript)

A few English corrections:”
“Data was” line 137 and “data were” in line 145; please use one form consistently
“Although the lack of evidence, falls in this specific context..” in line 35 should be rephrased 

Author Response

thank you very much for your comments
They have enabled us to improve the article
The changes are marked in the article

Round 2

Reviewer 1 Report

The authors addressed revision suggestions, and it's now ready for publication. 

Author Response

One of the Reviewers requested consistent use of “data were” or “data was” in the manuscript. The authors have made such consistence by using “data were” throughout the manuscript with only one exception remaining in the second sentence from the bottom of Abstract. 

authors change The influence of sociodemographic and clinical data was also checked - The influence of sociodemographic and clinical data were also checked